# Extracellular Proteome Analysis Shows the Abundance of Histidine Kinase Sensor Protein, DNA Helicase, Putative Lipoprotein Containing Peptidase M75 Domain and Peptidase C39 Domain Protein in *Leptospira interrogans* Grown in EMJH Medium

**DOI:** 10.3390/pathogens10070852

**Published:** 2021-07-06

**Authors:** Abhijit Sarma, Dhandapani Gunasekaran, Devasahayam Arokia Balaya Rex, Thoduvayil Sikha, Homen Phukan, Kumar Mangalaparthi Kiran, Sneha M. Pinto, Thottethodi Subrahmanya Keshava Prasad, Madathiparambil G. Madanan

**Affiliations:** 1Regional Medical Research Centre Port Blair, Indian Council of Medical Research, Port Blair 744103, Andaman and Nicobar Islands, India; abhijit.sarma2012@gmail.com (A.S.); gunasekaran.vpm1990@gmail.com (D.G.); sikha.tt@gmail.com (T.S.); biotechphukan16@gmail.com (H.P.); 2Department of Chemical Sciences, Ariel University, Ariel 70400, Israel; 3Center for Systems Biology and Molecular Medicine, Yenepoya (Deemed to be University), Mangalore 575018, India; rexprem@yenepoya.edu.in (D.A.B.R.); sneha.mp@gmail.com (S.M.P.); tskprasad@gmail.com (T.S.K.P.); 4Department of Pathology, Jawaharlal Institute of Postgraduate Medical Education and Research, Puducherry 605006, India; 5Institute of Bioinformatics, Bangalore 560066, India; mkirankumar45@gmail.com; 6NIMHANS-IOB Proteomics and Bioinformatics Laboratory, Neurobiology Research Centre, National Institute of Mental Health and Neuro Sciences, Bangalore 560029, India

**Keywords:** *Leptospira*, protein, extracellular, surface, secretory, pathogenic, proteomics

## Abstract

Leptospirosis is a re-emerging form of zoonosis that is caused by the spirochete pathogen *Leptospira*. Extracellular proteins play critical roles in the pathogenicity and survival of this pathogen in the host and environment. Extraction and analysis of extracellular proteins is a difficult task due to the abundance of enrichments like serum and bovine serum albumin in the culture medium, as is distinguishing them from the cellular proteins that may reach the analyte during extraction. In this study, extracellular proteins were separated as secretory proteins from the culture supernatant and surface proteins were separated during the washing of the cell pellet. The proteins identified were sorted based on the proportion of the cellular fractions and the extracellular fractions. The results showed the identification of 56 extracellular proteins, out of which 19 were exclusively extracellular. For those proteins, the difference in quantity with respect to their presence within the cell was found to be up to 1770-fold. Further, bioinformatics analysis elucidated characteristics and functions of the identified proteins. Orthologs of extracellular proteins in various *Leptospira* species were found to be closely related among different pathogenic forms. In addition to the identification of extracellular proteins, this study put forward a method for the extraction and identification of extracellular proteins.

## 1. Introduction

Leptospirosis, the zoonotic disease once confined to posing a risk during agricultural activities, has been re-emerging due to increasing urbanization and slum areas that have increased the reservoir rodent population [1]. The increase in outbreaks during floods has been due to water getting contaminated with the urine from rats and several other domestic and wild animals that spread out during the floods. Humans exposed to such water obtain the infection through cuts, wounds, and abrasions on the skin. Symptoms of leptospirosis appear after two weeks of infection and progress rapidly. Early diagnosis and treatment form the most important strategy to avoid complications and the loss of life due to leptospirosis. Alternatively, the development of efficient vaccines and drugs is also necessary. Rational development of diagnostic tools and vaccines requires a clear understanding of the molecules used as candidates or targets for this purpose [2]. Secretory proteins are a group of molecules from which better molecules for an anti-leptospiral strategy can be obtained because of their biological significance concerning pathogenesis and host-pathogen interactions [3]. Many pathogens secrete toxins, proteases, and a range of extracellular enzymes like lipase, fibrinolysin, hyaluronidase, protease, elastase, etc. [4,5,6].

It has been shown that there are many protease activities that can degrade the extracellular matrix and plasma proteins to contribute to the process of infection and pathogenesis in the culture supernatant of *L. interrogans* [7]. One of the earliest studies indicated the secretion of 60 kDa hemolysin, which was found to be inducing proinflammatory cytokines through the Toll-like receptor 2-and 4-mediated JNK and NF-κB signaling pathways [8,9]. Hemolysin and sphingomyelinase activity has been detected in the culture fluids of several strains of pathogenic *Leptospira* with specific antiserums [10,11,12]. Many immunoreactive proteins have been reported from acute and convalescent-phase sera of leptospirosis patients [13,14]. The *Staphylococcus aureus* secretes a variety of immune evasion molecules, including proteases that cleave components of the innate immune system and also disrupt the integrity of the extracellular matrix. The secretory proteins of *S. aureus* can activate host zymogens that target host-specific defense and inhibit the anti-bacterial functions of the host, thereby enhancing the chances of pathogen survival in the host [15]. A secreted protein kinase YpkA of *Yersinia pseudotuberculosis* is involved in pathogenicity by interfering with the signal transduction pathways of the host cell [16]. The *Coxiella burnetii* secretes Coxiella serine/threonine-protein kinase (CstK) that functions as a bacterial effector protein and assists in the biogenesis of parasitophorous vacuole and replication of the intracellular pathogen by interacting with the host protein TBC1D5 [17]. The involvement of secretory proteins in invasion and pathogenesis is also well established in many parasite infections. In *Toxoplasma gondii*, invasion and replication in the host is facilitated by secretory proteins by the modification of host cellular factors [18], which are governed by the parasite proteins secreted from their secretory organelles like rhoptries and dense granules [19].

Secretory proteins are good diagnostic targets during infection and can be a direct indicator of pathogen load in the host [20,21]. However, the difficulty of their isolation and purification of secretory proteins from biological samples, as well as culture supernatants, has restricted the research and exploration of their use in anti-leptospirosis strategies. The main reason for this is that many forms of culture media are very rich in proteins and are used as forms of enrichment in the medium, which makes the isolation of secretory proteins difficult. For example, the isolation of secretory proteins from culture supernatant containing abundant Bovine Serum Albumin (BSA) (10 mg/mL) present in the Ellinghausen-McCullough-Johnson-Harris (EMJH) medium, used to culture *Leptospira*, is a challenging process. In this context, a multipronged approach coupled with the enrichment of secretory proteins from culture supernatant, its analysis using high throughput Liquid Chromatography with tandem Mass Spectrometry (LC-MS/MS) based proteomics, and the use of bioinformatics tools for the selection of candidate proteins from the supernatant of *Leptospira* culture in complete EMJH medium was attempted. Previously, Triton X-114 fractionation coupled with LC-MS/MS-based proteomics to determine OMPs of *L. interrogans* [22] was used to compare the cellular portion with the extracellular proteins from the wash and supernatant of the culture identified through LC-MS/MS in this study. This article describes the enrichment of secretory proteins from the EMJH culture supernatant and surface proteins from the wash fraction and the proteomic and bioinformatics analysis of the results.

## 2. Results

LC-MS/MS analysis of TritonX-114 fractions of *L. interrogans* serovar Copenhageni strain Fiorcruz L1-130 and subsequent identification yielded 2425 proteins in aqueous fractions, 2646 protein in detergents, and 1684 proteins in a pellet that comprised 2957 unique proteins reported under the ProteomeXchange dataset identifiers PXD009050 and PXD016204 [22]. The analysis of wash and supernatant identified 837 proteins in wash 851 proteins in the supernatant comprising 1176 unique proteins (Figure 1A). The proteomics dataset is available through ProteomeXchange with the identifier PXD026044.

The abundance of secretory proteins across aqueous, detergent, pellet, wash, and supernatant fractions was calculated based on Intensity-Based Absolute Quantification (iBAQ) values (Table 1 and Appendix A). A protein was considered abundant in a fraction if the average iBAQ value of the protein from three replication was at least 1.5-fold higher than the average iBAQ value of the protein totaled out of all other fractions. This showed that 56 proteins were abundant out of 1176 proteins identified in the wash and supernatant together. Out of the 56 proteins, 19 were not found in cellular fractions (aqueous, detergent, and pellet proteins together). Out of the 19 proteins, six were found only in the wash fraction (surface), nine were found exclusively in the supernatant (secretory), and four were found in both with an abundance in the supernatant (Figure 1B).

### 2.1. Bioinformatics Analysis of Secretory Proteins

#### 2.1.1. Prediction of Subcellular Location of Extracellular Proteins Identified in *Leptospira*

Our results showed that CELLO v.2.5 predicted that 21.4% of the extracellular proteins would be same, while PSORTb predicted 12.5% and BUSCA predicted 8.9% to be extracellular (Figure 2). Most of the predictions made by CELLO v.2.5 involved cytoplasmic (41%) and Outer Membrane Protein (OMP) proportions (26.8%) In the case of PSORTb, the predictions were that it was cytoplasmic (26.8%) and unknown (35.7%) while the BUSCA predicted that cytoplasmic (53.6%) and inner membrane proteins (35.7%) would be a major portion of *Leptospira* (Table 1).

#### 2.1.2. Prediction of Protein Function and Pathogenic Nature

Identification of the role of the proteins identified as extracellular is important to assess their usefulness in an anti-leptospirosis strategy. The KEGG database showed seven classes of functional groups in the 56 proteins (Figure 3A). Four proteins were found under a metabolism that includes the Amino acid metabolism (1), Carbohydrate metabolism (1), Energy metabolism (1), and Metabolism of cofactors and vitamins (1). In the Cellular Processes under the Cellular community–prokaryotes, in the Genetic Information Processing under folding, sorting and degradation, and in Environmental Information Processing under Membrane transport one protein each category. The BRITE hierarchy showed 8 proteins which are distributed under the Protein families; genetic information processing 3, metabolism 3, signaling and cellular processes 2 proteins were present. Under Genetic Information Processing, two Replication and repair proteins and one Translation activity were present. There was one Unclassified signaling and cellular processes protein that is not included in any pathway or BRITE.

The severity of an infection caused by a pathogen is determined by the virulent and pathogenic proteins present in an organism. The MP3 online server predicted that out of 56 proteins, 32 were pathogenic/virulent and 24 were nonpathogenic (Figure 3B). The abundant 37 proteins that were found to be cellular included 20 pathogenic proteins and 17 nonpathogenic proteins. Out of the 19 exclusive proteins found in wash and supernatant, 12 were pathogenic and seven were nonpathogenic. There were eight pathogenic proteins out of 10 exclusive proteins found in the wash.

These results showed that the proteins determined to be secretory may provide a balance between the pathogenic and non-pathogenic proteins which progressively determine the *Leptospira* based on the environmental conditions.

#### 2.1.3. Identification of Orthologous Proteins

The NCBI protein-protein BLAST (blastp) retrieved orthologous proteins of 56 secretory proteins. The ‘query cover’ and ‘identity’ values were used to analyze protein similarity with the query sequence, which was the secretory protein of *L. interrogans*. The *Leptospira* species represented in the BLAST result were arranged according to the P1, P2, S1, and S2 groups, and the same sequence was shown of a phylogenetic tree constructed with the ppk gene sequences in Figure 4 as from an earlier study that describes 64 species of *Leptospira* [23]. The result showed the highest coverage and identity of 26 extracellular proteins with 17 species of pathogenic *Leptospira* (Figure 4). The intermediate, as well as saprophytic species, showed less than 50% identity though having good coverage against the query sequences.

#### 2.1.4. Determination of Protein Interactions

As a representative study, four proteins: Histidine kinase sensor protein (LIC_11528), DNA helicase (LIC_11624), Putative lipoprotein containing peptidase M75 domain (LIC_10713), and Peptidase C39 domain protein (LIC_10511) showed an abundance of 1771-, 125-, 79-, and 76-times, respectively, in extracellular samples that were subjected to string analysis to find inter-correlation among the various proteins within the *Leptospira* group (Figure 5). The C39 peptidase group was found to be associated with LIC 10510 and LIC 10512 where no functional analysis was interpreted. Also, putative lipoprotein (Imelysin domain) was predicted to associate with gene encoding proteins i.e., LIC 10712, LIC 11466, and LIC 10711, thereby resembling a neighborhood and co-occurrence among the species. Further, histidine kinase sensory proteins were found to be associated with other chemotaxis response regulator proteins. This association was also determined by neighborhood relationships within the species. Further, DNA helicase was found to be involved with uvrB, mutL, recQ, recA, pheT, polA, and lig proteins. These associations were determined experimentally with DNA helicase.

## 3. Discussion

Bacteria secrete a wide variety of proteins, enabling them to respond to their environment. These extracellular proteins have a diverse functional role such as degradation of substrates, response to environmental stimulus, migration, genetic exchange, feeding, ion-capturing, and sociobiological aspects like quorum sensing, biofilm formation, and host-pathogen interaction [24,25]. In this context, identification and characterization of extracellular proteins of *Leptospira* help in elucidating the functions of proteins that lead to pathogenesis and survival of the pathogen in the host. This study was conducted with the same sample that describes cellular proteins. Identification of proteins that are exclusively present in extracellular samples like wash and supernatant in spite of high-resolution analysis cellular fractions, which were separately analyzed in three (aqueous, detergent, and pellet) fractions, underlines the quality of extracellular protein preparations. Similarly, the exclusive proteins found within wash and supernatant samples showed the discriminatory power of the preparations to distinguish between surface and secretory groups in extracellular proteins. Out of 1176 unique proteins identified from wash and supernatant samples, 56 proteins were found with >1.5-fold abundance that indicates that 4.76% were extracellular proteins. The 19 abundant proteins found exclusively in the extracellular group showed a 1.61% rate of proteins identified in the wash and supernatant. The surface proteins were extracted in PBS containing 5 mM MgCl2 that was easily detachable and also had the unique presence of six proteins without shedding out into the supernatant, which showed its sufficiently good binding on the surface of *Leptospira*. It is also worth noting that 57% of the extracellular proteins found were pathogenic in nature, as was predicted by MP3 tool with a 27.9% rate (826 pathogenic proteins out of the 2957 proteins identified) in the whole proteome of *Leptospira* [22]. Even though some of these proteins were found in lower quantities, the possibility of upregulation of the proteins under pathogenic conditions cannot be ignored. This underlines the significance of identification of these extracellular proteins.

The bioinformatics prediction of extracellular proteins using online tools did not achieve any appreciable level, as the true prediction with respect to 56 extracellular proteins made by CELLO v.2.5, PSORTb, and BUSCA were 21.4%, 12.5%, and 8.9%, respectively. Similarly, five proteins of the extra-cellular proteins were identified as extracellular proteins predicted in a previous report [25]. This shows the need for further improvements in the algorithms to predict extracellular proteins of *Leptospira*.

Though the species was arranged against a preexisting phylogenetic hierarchy, based on ppk gene sequences, the data matched with the clusters and sequence of species. This shows that the extracellular proteins can discriminate between *Leptospira* species and the unique extracellular proteins may have key functions in invasion, pathogenesis, or survival of the organism in the host. These can be used for diagnostic applications and identification of *Leptospira* species. The orthologous proteins in other species of *Leptospira* showed that all the 17 pathogenic proteins were closely related to the *L. interrogans* with respect to the 26 proteins with >90% coverage and >75% identity among the species. Three proteins: the WP_000141830.1 a Multispecies preprotein translocase subunit YajC, the WP_000587664.1 as a VOC family protein, and the WP_000658301.1 as a DNA starvation/stationary phase protection protein were found to be more closely related within the pathogenic forms. With respect to the preprotein translocase subunit YajC, it was reported that mice vaccinated with the yajC of *Brucella abortus* showed immune responses to YajC [26].

With an abundance of 1700-times in wash with respect to cellular and supernatant fraction, histidine kinase sensor protein (LIC_11528) was identified as the most abundant and pathogenic protein identified on the surface of *Leptospira* in this study. The gene ontology was predicted to carry functions like signal transduction, and phosphorylation and molecular functions like phosphorylase sensor kinase activity and transferase activity by transferring phosphorus-containing groups. This shows that the protein is a two-component system with a histidine protein kinase (HPKs) and a response regulator protein [27]. The phosphorylation can induce conformational changes in the regulatory domain, resulting in the activation of the associated domain that affects the response. This shows that bacterial two-component pathways can control a dazzling array of functions like cell division, virulence, antibiotic resistance, response to environmental stress, sporulation, metabolite fixation and utilization, and taxis [28]. In addition, HPKs are unique signal transducers that are not common in animals, indicating this molecule can be a good target for an anti-leptospiral strategy [29]. Further, the STRING analysis identified the interaction of this domain with a cheY protein resembling the bacterial adaptions to the environment through the activation of specific sensory receptors along with signal processing proteins [30]. Apart from these properties, LIC_11528 is an ortholog of LA_2421 of *L. interrogans* serogroup Icterohaemorrhagiae serovar Lai and was found to be associated with the chemotaxis and signal transduction system [31]. Chemotaxis is one of the important mechanisms that drive a pathogen towards its target organs [32].

The second abundant protein was DNA helicase (LIC_11624,) which was found to be 38-fold greater on the surface and 87-fold greater in the supernatant than its entire quantity in the cell. This was found to be nonpathogenic in MP3 and the gene ontology showed molecular functions like DNA helicase activity, ATP binding, DNA binding, and hydrolase activity. The STRING analysis showed the association with uvr A and uvr B types of DNA repair gene homologs, which are involved in repairing of DNA that is damaged due to stress factors such as ultraviolet light [33,34]. DNA helicase is also associated with proteins like the DNA mismatch repair protein MutL, which determines the mismatched provoked excision step [35] and the RecQ helicase, which are widely conserved in bacteria [36] and helps in unwinding complementary strands of DNA required for the proper repair of DNA damage. However, the function of this DNA helicase as an extracellular entity is not yet known.

The putative lipoprotein containing peptidase M75 domain (LIC_10713) was found to be 78-times more abundant in the supernatant as a secretory molecule. The protein contains an Imelysin-like domain with a GxHxxE signature. This domain was distributed widely in bacteria and was found to be involved in iron transport [37]. This protein showed 100% Query Cover and 97.73% identity LruB, which was found to be playing a significant role in human and equine recurrent uveitis as well as antibodies against the protein are identified in patients diagnosed with Fuchs uveitis [38,39,40]. It is also found that the in vitro growth of *Leptospira* was significantly reduced when LruB is inactivated [41].

The gene LIC_10511 encoding the protein C39 peptidase, which has been found to be 75-fold abundant in the supernatant, was reported to be an endo-peptidase family that mostly serves as ABC transporters along with the translocation of the mature bacteriocin across the cytoplasmic membrane [42]. This protein was also abundantly present in the supernatant resembling type III secreted effectors (T3SEs), type IV secretion (T4SS), and a type VI secretion system [43,44,45]. The Type VI secretion system was widely reported in *E. coli*, which was found to be directly associated to pathogenesis leading to macrophage survival, which can further lead to events of lateral gene transfer [46,47]. This interconnected mechanism allows the bacteria to participate in metal uptake and provide an advantage during bacteria-bacteria competition. Further, it also allows the bacteria to widely deliver the effector toxic proteins directly into neighboring cells [48].

Apart from these four proteins, other proteins also showed a significant rise in quantity as secretory molecules as compared to their presence in aqueous, detergent, and pellet fractions. These proteins were found to carry functions like cell-cell signaling, determining nutritional requirement, stress response, external specific stimuli, and homeostasis, which were interpreted by InterProScan 5. Further, significant domain class proteins like chaperone proteins, VOC family, SGNH hydrolase, ComF, and ATP binding cassette proteins were also identified from our study. Previous studies reported the presence of these domains in *Leptospira* determining the functions like survival under stress conditions, cell-cell signaling, and binding to membrane receptors [49,50,51].

## 4. Materials and Methods

### 4.1. Study Design

The *Leptospira* culture in complete EMJH medium at standard culture condition was used for a complete subcellular proteomic analysis using Triton X-114, as shown in Figure 6. The study comprised of two parts: (1) subcellular proteome which includes all the fractions of Triton X-114 fractionation considered as cellular proteins; and (2) the extracellular proteins obtained from a wash of the *Leptospira* pellet (surface) and proteins enriched from the culture supernatant (secreted). The total amount of each protein from Triton X-114 fractions identified in part one was added together and considered as a cellular portion of the protein in contrast to the number of the same extracellular proteins found, which is part two.

### 4.2. Leptospira Strain and Culture

*L. interrogans* Copenhageni stain Fiocruz L1-130 was obtained from the repository of the ICMR-Regional Medical Research Centre, Port Blair, India. This is a WHO Collaborating Center for diagnosis, reference, research, and training in leptospirosis. *Leptospira* were cultured in EMJH medium supplemented with 1% bovine serum albumin at 30 °C with intermittent checking for contamination and growth. Afterwards, they were harvested at the mid-log phase for further protein extraction.

### 4.3. Enrichment of Surface and Extracellular/Secreted Protein

The mid-log phase culture of *Leptospira* was centrifuged at 2500× *g* for 30 min at 4 °C to obtain the culture supernatant for the separation of secretory protein. This supernatant was further centrifuged at 6000× *g* for 30 min to remove any *Leptospira* left in the medium and the supernatant was again centrifuged at 12,000× *g* for 30 min. The three-step centrifugation was to avoid tight packing and rupture of *Leptospira* while pelleting. This supernatant was used to separate secretory/extracellular proteins.

### 4.4. Extraction of Extracellular/Secreted Proteins

The secretory proteins present in low abundance in comparison with the BSA or serum proteins. Enrichment of secretory proteins was carried out using ProteoMiner™ Bio-Rad (Hercules, CA, USA) protein enrichment technology based on binding of proteins to a library of combinatorial peptide ligands that act as unique binders for proteins [52,53].

The supernatant was dialyzed against PBS (containing 150 mM NaCl, pH 7.4) to facilitate optimum binding condition to ProteoMiner™. Slurry from ProteoMiner™ Large-Capacity column (100 μL settled beads) was washed two times with 1 mL of PBS and added to 100 mL of the supernatant and allowed to bind overnight (>8 h) under shaking at 4 °C. After binding, the beads were allowed to settle and we removed the clear volume of supernatant, repacked in the ProteoMiner™ column, and carried out 2 × 100 µL washes with PBS. The elution 2 × 20 µL was made using Elution Reagent (8 M urea, 2% CHAPS) supplied by the manufacturer. The eluted secretory protein was subjected to quantification, electrophoretic characterization, and trypsin digestion to obtain peptides and further high-resolution LC-MS/MS based proteomics.

### 4.5. Extraction of Surface Proteins

The pellet obtained after separation of supernatant was washed 3 × with PBS containing 5 mM MgCl2 and we collected the wash supernatant by centrifuging the leptospires at 2500× *g* for 5 min at room temperature. The wash was again centrifuged at 12,000× *g* for 30 min to remove any trapped leptospires and the supernatant was designated as a ‘Wash fraction’ that contains washable surface proteins of Leptospira and used for further processing to carry out LC-MS/MS.

### 4.6. Triton X-114 Extraction

The *Leptospira* pellet obtained after wash was used for Triton X-114 fractionation as described earlier [22]. The extraction buffer containing 10 mM Tris (pH 8) carrying 1% Triton X-114 and 150 mM NaCl at 4 °C at the rate of 1 mL of extraction buffer per amount of pellet derived from a 25 mL mid-log phase culture was used for extraction. The extract was centrifuged at 12,000× *g* for 30 min at 4 °C and the pellet was saved as a ‘pellet fraction’ and the supernatant was used for phase separation. The Triton X-114 concentration of the supernatant was increased to 2% by the addition of an adequate amount of Triton x-114, depending on the volume, mixed well, and incubated at 37 °C for 1 h for phase separation, then it was subsequently centrifuged at 1500× *g* for 5 min to separate the upper aqueous phase from the lower detergent phase. The undissolved proteins from the pellet from the TritonX-114 extraction step, which contained a cytoplasmic cylinder, were further extracted using a buffer containing 10 mM Tris-Cl (pH8), 8 M urea, 4 mM dithiothreitol, and 1% sodium dodecyl sulfate. Following centrifugation at 12,000× *g* for 30 min at 4 °C, the supernatant was used as the pellet fraction. Similar fractions of all four replicated were polled and the protein concentrations were estimated using the BCA method in the aqueous, detergent, and pellet fractions, which were then stored at −20 °C. This protein was used for mass spectrometry. Data from three replications of the same kind was used for further analysis.

### 4.7. Mass Spectrometry Analysis

#### 4.7.1. In-Solution Digestion

Wash and supernatant samples protein concentrations were estimated by using the BCA (Bicinchoninic Acid) Protein Assay (Pierce™ BCA Protein Assay Kit), and the protein amount was reconfirmed visually resolving on a 10% SDS-polyacrylamide gel electrophoresis (PAGE) gel. Based on the protein concentrations, a quantity equivalent to 250 g of protein was taken from wash and supernatant samples. Further samples were reduced with 10 mM dithiothreitol (DTT) and alkylated with 20 mM iodoacetamide (IAA). Prior to trypsin digestion, the lysate was precipitated with acetone to remove sodium dodecyl sulfate (SDS) to form the protein sample. The protein was digested with trypsin (1:20) (modified sequencing grade; Promega, Madison, WI, USA) at 37 °C for 16 h. The peptides were dried overnight in SpeedVac and stored at −20 °C.

#### 4.7.2. Basic pH RPLC Based Fractionation

Lyophilized peptides were subjected to basic pH reverse phase chromatography (bRPLC) fractionation. The samples were reconstituted in 1 mL of 10 mM Triethylammonium bicarbonate (TEABC) and separated on an XBridge C18 column (Waters Corporation, Milford, MA, USA; 130, 5 m, 250 × 4.6 mm) attached to a Hitachi LaChrom Elite HPLC system over 120 min using a linear gradient increase from 5% to 100% of 10 mM TEABC with 90% acetonitrile. Initially, 96 fractions were collected, which were then concatenated to 6 fractions and dried before desalting with C18 cartridges. Desalted peptides were vacuum dried and stored in a deep freezer at −80 °C prior to LC-MS/MS analysis.

#### 4.7.3. LC-MS/MS Analysis

The tryptic peptides from bRPLC fractionation were analyzed on a Thermo Fischer Scientific Orbitrap Fusion Tribrid mass spectrometer (Thermo Fischer Scientific, Bremen, Germany) connected with an Easy-nLC-1200 nanoflow liquid chromatography system (Thermo Fischer Scientific). The lyophilized peptides were reconstituted in 0.1% formic acid and loaded onto a 2 cm trap column (nanoViper, 3 µm C18 Aq) (Thermo Fisher Scientific). Peptides were separated using a 15 cm analytical column (nanoViper, 75 µm silica capillary, 2 µm C18 Aq) at a flow rate of 300 nl/min. For data-dependent acquisition, solvent gradients were set as the linear gradient of 5–35% solvent B (80% acetonitrile in 0.1% formic acid) over 90 min through 120-min run time. MS analysis was carried out at a scan range of 400–1600 *m*/*z* mass range (120,000 mass resolutions at 200 *m*/*z*). The maximum injection time was 10 ms. For MS/MS analysis, data were acquired at top speed mode with 3 s cycles and subjected to a higher collision energy dissociation with 32% normalized collision energy. MS/MS scans were carried out at a range of 100–1600 *m*/*z* using Orbitrap mass analyzer at a resolution of 30,000 at 200 *m*/*z*. The maximum injection time was 200 ms.

#### 4.7.4. MS/MS Data Analysis

Mass spectrometry-derived data were searched against the *L. interrogans* serogroup Icterohaemorrhagiae serovar Copenhageni (strain Fiocruz L1-130) reference protein database obtained from NCBI (3667 protein entries), with common contaminants added to the protein database (115 contaminants entries). The mass spectrometry data was analyzed with Mascot (versions 2.5.1; Matrix Science, London, UK) and SEQUEST-HT search algorithms in the Proteome Discoverer software suite, version 2.2 (PD 2.2) (Thermo Fischer Scientific, Bremen, Germany). The search parameters used were: (a) trypsin as the proteolytic enzyme (with up to one missed cleavage); (b) fragment mass error tolerance of 0.05 Da; (c) peptide mass error tolerance of 10 ppm; (d) oxidation of methionine as a variable modification; (e) carbamidomethylation of cysteine as a fixed modification. A false discovery rate (FDR) was set to 1% at PSM and peptide levels. The iBAQ (Intensity Based Absolute Quantification) value was generated using the iBAQ algorithm that estimates the relative abundance of the proteins within each sample [54].

#### 4.7.5. Data Availability

The proteomics data of these mass spectrometry analyses have been deposited to the ProteomeXchange Consortium via the PRIDE [55], the partner repository with the dataset identifier.

#### 4.7.6. Bioinformatics Analysis

##### Sub-Cellular Localization

To predict the sub-cellular location of secreted proteins, online tools like CELLO v.2.5 [56,57] (cello.life.nctu.edu.tw/, accessed on 5 March 2020) and PSORTb version 3.0.2 [58] tool (https://www.psort.org/psortb/, accessed on 9 March 2020) were used. Additionally, a web-server BUSCA (Bologna Unified Subcellular Component Annotator, http://busca.biocomp.unibo.it/, accessed on 19 April 2021) which integrates different tools (DeepSig, TPpred3, PredGPI, BetAware, and ENSEMBLE3.0) to predict localization-related protein features as well as tools like BaCelLo, MemLoci, and SChloro for discriminating subcellular localization of both globular and membrane proteins while predicting subcellular localization [59].

##### Prediction of Protein Function and Pathogenic Nature

Functional annotation of the secretory protein is important to know the role of these proteins. Identification of functions and metabolic pathways of these proteins were made using the KEGG database (https://www.genome.jp/kegg-bin/show_brite?lic, accessed on 28 February 2021) [60]. Similarly, virulence and pathogenicity are key determinants of the severity of infection caused by a pathogen. To identify virulent proteins, we used MP3 (http://metagenomics.iiserb.ac.in/mp3/application.php, accessed on 12 April 2020) [61]. This tool is an SVM-based method to characterize the pathogenic proteins from the non-pathogenic ones.

##### Identification of Orthologous Proteins

A search for orthologous proteins of extracellular within 64 species of *Leptospira* was carried out using algorithm blastp (protein-protein BLAST) under online NCBI BLAST search at default parameters to retrieve top 1000 hits. Orthologous proteins of the highest score from each *Leptospira* species, irrespective of strains, were selected along with their ‘query cover’ and ‘identity’ values with respect to the query sequence.

##### Analysis of Interacting Proteins

To predict the protein-protein interaction among the species of *Leptospira*, we used String database (http://version10.string-db.org/, accessed on 21 May 2021) version 10. Under the search option, we entered the unique protein uniport ID and selected the autodetect option [62].

## 5. Conclusions

This study aimed to identify extracellular proteins of *L. interrogans* from protein-rich EMJH medium. It shows that the surface and secretory proteins can be easily identified with reference to the cellular proteins in quantitative terms. The extraction method was found to be easy, rational, and justified with the identification of exclusive molecules and significant times of abundance with respect to the cellular fraction of the proteins, though it was analyzed at higher resolutions due to three Triton X-114 fractions. Identification of pathogenic proteins and the correlation with pathogenic species shows the significance of the identified proteins. Similarly, a huge number (57%) of pathogenic proteins present as secretory molecules also highlight the significance of extracellular proteins. These key molecules identified can implement various functions like nutrient acquisition, cell-cell communication, detoxification of environment, and attaching to potential inhibitors. In this regard, the article presents an efficient method for extraction and analysis of extracellular proteins for other organisms too, as well as identification of extracellular proteome of *L. interrogans*.

## Figures and Tables

**Figure 1 pathogens-10-00852-f001:**
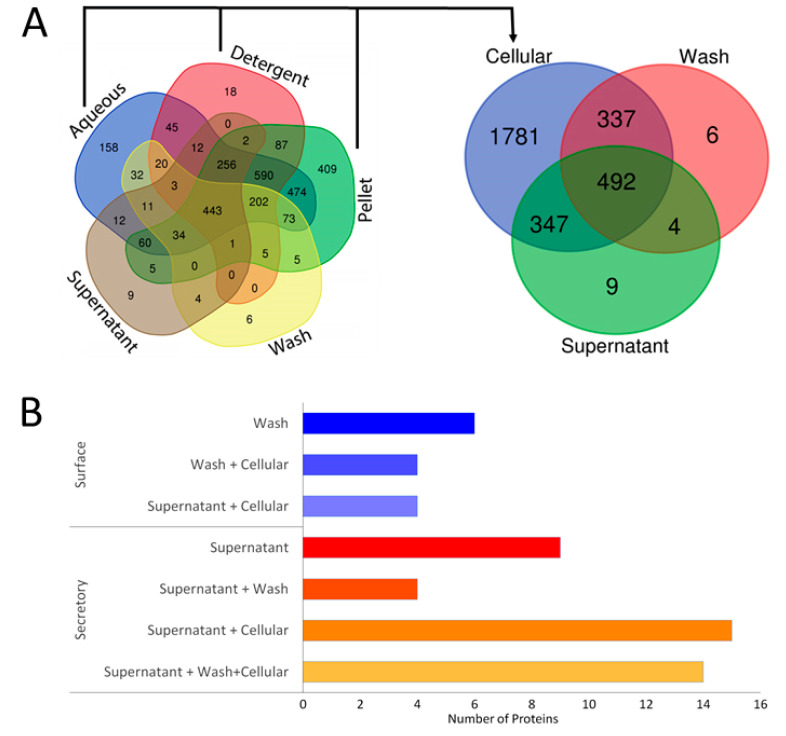
Mass spectrometry of leptospiral proteins. The Venn diagram shows the distribution of proteins identified from the *Leptospira* (**A**) culture pellet that was fractionated using Triton X-114 (aqueous, detergent, and pellet), the culture ‘supernatant’ obtained after centrifugation and pool of three washes, with PBS containing 5 mM MgCl2, of the pellet designated as ‘wash’ as well as the merged Triton X-114 fractions designated as the ‘cellular’ protein. (**B**) Shows iBAQ based abundance of 56 unique extracellular proteins distributed in the wash (surface) and supernatant (secreted).

**Figure 2 pathogens-10-00852-f002:**
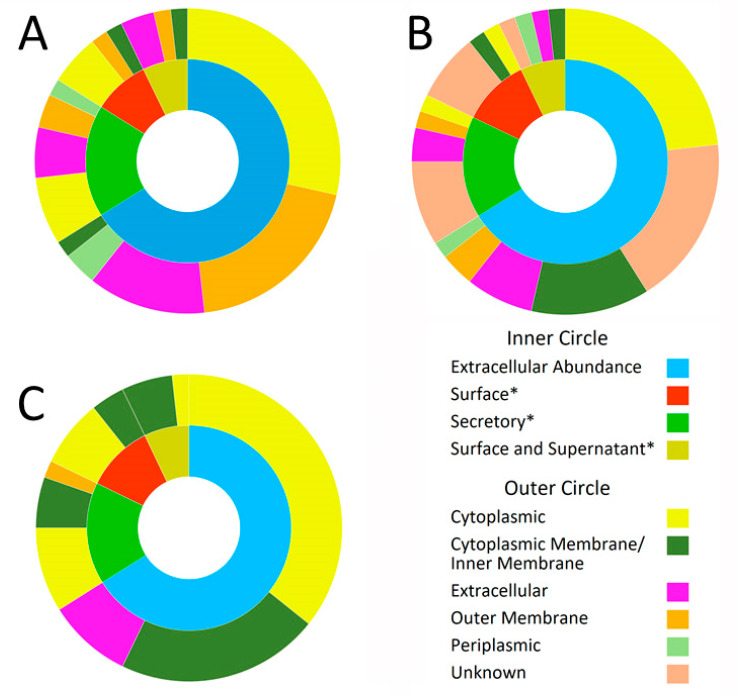
Prediction of extracellular proteins. Distribution of a number of extracellular proteins identified in the abundance-based groups of wash and supernatant of *Leptospira* culture in EMJH medium (the inner circle) into the bioinformatically predicted locations of the proteins (outer circle). The ‘*’ represents proteins identified exclusively from the indicated fraction. Part (**A**) shows the distribution of CELLO v.2.5 predicted subcellular locations, (**B**) shows PSORTb predicted subcellular locations, and (**C**) BUSCA predicted subcellular locations of extracellular proteins identified from wash and supernatant (Appendix A).

**Figure 3 pathogens-10-00852-f003:**
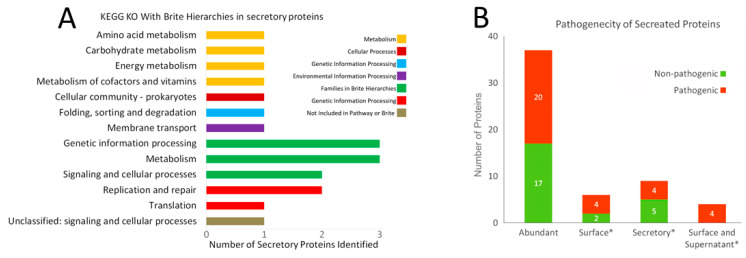
Functional characteristics of extracellular proteins. The bar graph (**A**) shows the functional annotation of 56 extracellular proteins based on the KEGG database that shows seven families of proteins and the number of class proteins in each family. The details are shown in the Appendix A. Part (**B**) shows pathogenic/virulent proteins identified using MP3 online server represented as abundance-based groups where ‘Abundant’ proteins are those proteins found to be extracellular due to indicating a >1.5 times higher amount present in the wash (surface) and supernatant (secretory). * indicates proteins that were exclusively found on the surface and/or as secretory and were not found in any cellular fractions.

**Figure 4 pathogens-10-00852-f004:**
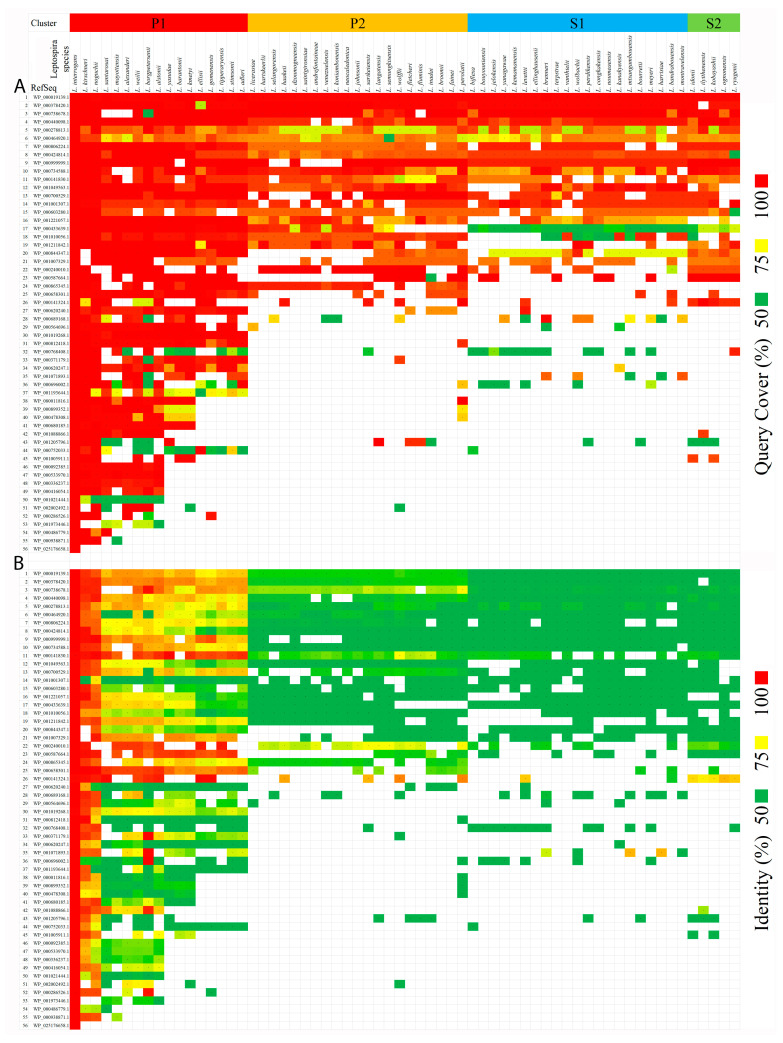
Analysis of orthologous proteins from 64 *Leptospira* species. The figure shows ‘query cover’ (**A**) and ‘identity’ (**B**) of orthologous proteins found in 64 species of *Leptospira* in an NCBI–BLAST search against 56 extracellular proteins of *L. interrogans*. The proteins represented as RefSeq were arranged in the highest to the lowest order of the sum of query cover and identity values that each protein scored from those *Leptospira* species it was identified as being part of. The *Leptospira* species were arranged based on their phylogenetic hierarchy and based on ppk gene sequences.

**Figure 5 pathogens-10-00852-f005:**
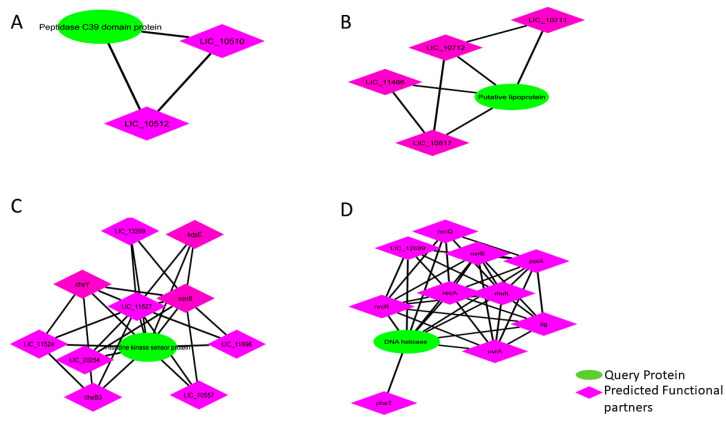
STRING based analysis of extracellular proteins. Each group of interactome depicts the interaction among the proteins within the *Leptospira* group: (**A**) Peptidase_C39 domain protein; (**B**) Putative lipoprotein; (**C**) Histidine kinase sensor protein; (**D**) DNA helicase.

**Figure 6 pathogens-10-00852-f006:**
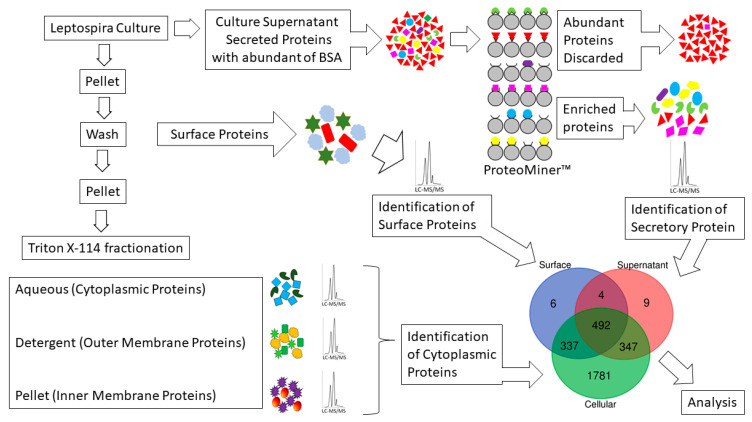
Proteomics workflow to identify extracellular proteins. The study design shows Triton X-114 fractionation of cellular proteins, extraction of surface proteins, and ProteoMiner™ (Bio-Rad) based enrichment of secreted proteins from *Leptospira* culture supernatant for LC-MS/MS analysis.

**Table 1 pathogens-10-00852-t001:** Abundance based sorting of extracellular proteins using iBAQ values. Selection of 56 extracellular out of 1195 proteins identified through LC-MS/MS in wash and supernatant of *Leptospira* grown in EMJH medium. The values shown are average of normalized iBAQ values from three replicates. The abundance is indicated as exclusively present in wash (Ex. W), exclusively present in supernatant (Ex. S), and present only in wash, while supernatant (Ex. EC) and those found in cellular fractions also are shown in values indicating a fold of abundance as being extracellular. The protein is not identified in the fraction (-). The proteins found abundant in the cellular fractions are submitted as Appendix A.

Sl. No.	Accession	Gene Names (Ordered Locus)	Description	Total Value of Triton X-114 Fractions (C)	Wash (W) (Surface)	Supernatant (S) (Secretory)	Abundance of Extracellular Proteins (EC) = (W+S)/C
1	WP_000286526.1		hypothetical protein	-	0.1	-	Ex. W
2	WP_000424814.1		ComF family protein	-	8.4	-	Ex. W
3	WP_000486779.1	LIC_10697	hypothetical protein	-	2.2	-	Ex. W
4	WP_000700529.1		hypothetical protein	-	0.5	-	Ex. W
5	WP_000865345.1	LIC_13053	fatty acid desaturase	-	0.3	-	Ex. W
6	WP_001973446.1		hypothetical protein	-	0.7	-	Ex. W
7	WP_000371179.1	LIC_13177	hypothetical protein	-	-	5.9	Ex. S
8	WP_000416054.1	LIC_11345	TonB-dependent siderophore receptor	-	-	0.04	Ex. S
9	WP_000738678.1	LIC_12988	lipase	-	-	14.3	Ex. S
10	WP_000812418.1	LIC_10645	hypothetical protein	-	-	1.4	Ex. S
11	WP_000844347.1	LIC_10346	SGNH/GDSL hydrolase family protein	-	-	1.2	Ex. S
12	WP_000899352.1		sphingomyelin phosphodiesterase	-	-	1.7	Ex. S
13	WP_001021444.1		DUF1563 domain-containing protein	-	-	18.2	Ex. S
14	WP_001088866.1	LIC_12191	NUDIX hydrolase	-	-	2.3	Ex. S
15	WP_025176658.1		hypothetical protein	-	-	17.3	Ex. S
16	WP_000336237.1	LIC_12715	DUF1561 domain-containing protein	-	0.7	3.2	Ex. EC
17	WP_000533970.1	LIC_12986	DUF1561 domain-containing protein	-	0.5	3.9	Ex. EC
18	WP_001193644.1		hypothetical protein	-	2.1	4.7	Ex. EC
19	WP_001211842.1	LIC_11088	di-heme enzyme	-	0.4	4.8	Ex. EC
20	WP_000433639.1	LIC_11528	PAS domain S-box protein	0.5	941.1	7.0	1771.4
21	WP_000378420.1	LIC_11624	ATPase AAA	21.5	813.4	1871.9	124.9
22	WP_001049563.1	LIC_10713	peptidase M75	23.5	11.8	1833.1	78.5
23	WP_001205796.1	LIC_10511	hypothetical protein	0.1	-	4.0	75.6
24	WP_000938871.1	LIC_20255	hypothetical protein	1.7	42.3	29.6	42.5
25	WP_000752033.1	LIC_10370	hypothetical protein	2.1	10.8	72.9	38.9
26	WP_001071893.1	LIC_10704	hypothetical protein	2.9	-	101.8	35.2
27	WP_000689168.1		SGNH/GDSL hydrolase family protein	2.0	-	55.3	28.0
28	WP_002002492.1	LIC_11904	hypothetical protein	3.3	-	86.1	26.3
29	WP_001010056.1		hypothetical protein	3.6	-	94.0	26.2
30	WP_000141324.1	LIC_11265	MULTISPECIES: DUF1858 domain-containing protein	5.4	-	139.8	25.9
31	WP_000620240.1	LIC_10371	hypothetical protein	12.1	2.4	246.8	20.7
32	WP_080011816.1		sphingomyelin phosphodiesterase	0.4	-	7.0	19.4
33	WP_000806224.1	LIC_13164	2-amino-4-hydroxy-6-hydroxymethyldihydropteridine diphosphokinase	11.1	1.0	206.8	18.6
34	WP_000768408.1	LIC_12891	hypothetical protein	2.2	1.6	35.9	16.7
35	WP_000680185.1	LIC_10365	DUF1565 domain-containing protein	0.6	1.3	7.6	14.7
36	WP_000620247.1	LIC_10373	hypothetical protein	4.6	2.5	65.8	14.7
37	WP_000564696.1	LIC_10183	hypothetical protein	0.8	11.0	-	14.4
38	WP_000999999.1	LIC_11240	ATP synthase subunit delta	205.7	0.7	2731.9	13.3
39	WP_000696002.1	LIC_13248	DUF1554 domain-containing protein	0.7	-	8.6	12.6
40	WP_000092385.1	LIC_12791	DUF1561 domain-containing protein	0.6	-	6.7	11.7
41	WP_000019139.1	LIC_12413	DUF115 domain-containing protein	3.4	-	38.3	11.2
42	WP_000603280.1	LIC_20247	lytic transglycosylase domain-containing protein	0.7	-	6.9	10.3
43	WP_000734588.1		hypothetical protein	11.2	-	92.3	8.2
44	WP_000440098.1		DUF2203 domain-containing protein	3.1	20.5	-	6.6
45	WP_000141830.1	LIC_12540	MULTISPECIES: preprotein translocase subunit YajC	1067.4	1534.6	4361.7	5.5
46	WP_001001307.1		hypothetical protein	0.5	2.4	-	5.2
47	WP_001019268.1	LIC_12755	MULTISPECIES: hypothetical protein	94.8	21.7	289.9	3.3
48	WP_001005911.1	LIC_10552	hybrid sensor histidine kinase/response regulator	0.8	-	2.7	3.2
49	WP_000587664.1		VOC family protein	198.7	35.6	428.8	2.3
50	WP_000658301.1		DNA starvation/stationary phase protection protein	33.3	54.1	22.1	2.3
51	WP_000278813.1		DUF192 domain-containing protein	2.0	2.9	1.2	2.1
52	WP_001221057.1	LIC_10711	hypothetical protein	5.1	8.7	-	1.7
53	WP_001007329.1	LIC_12437	hypothetical protein	94.4	8.0	151.1	1.7
54	WP_000464920.1		stage II sporulation protein E	2.6	-	4.1	1.6
55	WP_000478308.1		sphingomyelin phosphodiesterase	1.8	0.7	2.1	1.6
56	WP_000240010.1	LIC_11555	30S ribosomal protein S16	27.9	-	42.3	1.5

## Data Availability

The LC-MS/MS proteomics dataset is available through ProteomeXchange dataset identifiers PXD009050, PXD016204 and PXD026044.

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
