# Peer review of "Extracellular Proteome Analysis Shows the Abundance of Histidine Kinase Sensor Protein, DNA Helicase, Putative Lipoprotein Containing Peptidase M75 Domain and Peptidase C39 Domain Protein in Leptospira interrogans Grown in EMJH Medium"

_pathogens, 2021, doi:10.3390/pathogens10070852_

Round 1
Reviewer 1 Report
The work is well written; despite having complex scientific methods, it is written in an intelligible way.
A work on pathogenic Leptospira spp. is always important, because there is little information concerning these pathogens. There are few groups who work on this zoonosis.
This is one of the few scientific works, which reports the application of the LC-MS/MS to pathogenic bacteria of Leptospira genus to search for their extracellular proteins.
This last point is important as a basis for identifying targets for pharmacological therapies or also for identifying future ways for new vaccines.
Author Response
Reviewer’s Comments and Suggestions for Authors
The work is well written; despite having complex scientific methods, it is written in an intelligible way.
A work on pathogenic Leptospira spp. is always important, because there is little information concerning these pathogens. There are few groups who work on this zoonosis.
This is one of the few scientific works, which reports the application of the LC-MS/MS to pathogenic bacteria of Leptospira genus to search for their extracellular proteins.
This last point is important as a basis for identifying targets for pharmacological therapies or also for identifying future ways for new vaccines.
Author’s response: The authors thank the reviewer for their patience and time to review the manuscript. The authors their gratefulness to the reviewer for the recommendation and appreciation of their work.
Reviewer 2 Report
The paper is very interesting describing a new approach to isolation and extraction methods of surface and secretory leptospira proteins. The authors undertook the difficult analysis and differentiation of pathogenic and nonpathogenic leptospira proteins and the interactions between these proteins.
I would like to highlight the following very minor shortcomings: line 81- no description to the BSA abbreviation; lines 249, 275- minor errors of punctuation marks; lines 33 to 34 - stylistic errors.
In addition, I believe that the entire text should be corrected by a professional native speaker, because I have the impression that once the text flows and sometimes I read with resistance.
Author Response
Comments and Suggestions for Authors
- Reviewer’s comment: The paper is very interesting describing a new approach to isolation and extraction methods of surface and secretory Leptospira The authors undertook the difficult analysis and differentiation of pathogenic and nonpathogenic Leptospira proteins and the interactions between these proteins.
Author’s response: The authors thank the reviewer for their patience and time to review the manuscript, their constructive comments and appreciation.
- Reviewer’s comment: I would like to highlight the following very minor shortcomings: line 81- no description to the BSA abbreviation; lines 249, 275- minor errors of punctuation marks; lines 33 to 34 - stylistic errors.
Author’s response: The authors added abbreviation for BSA and thoroughly checked the entire document for errors and punctuation including those mentioned and corrections are made.
- Reviewer’s comment: In addition, I believe that the entire text should be corrected by a professional native speaker, because I have the impression that once the text flows and sometimes I read with resistance.
Author’s response: The authors contacted experts in English language and made a thorough revision.
Reviewer 3 Report
The paper submitted by Sarma and colleagues showed the identification of protein in Leptospira interrogans by external proteome analysis.
The paper is very interesting and gives important information on leptospirosis.
I suggest changing the title, it is too long, confused, and full of brackets.
Need references in the sentences from line 40 to line 50. also lines 75-80. Besides in the introduction, there are too long sentences without references.
Please define LC-MS/MS the first time in the text.
L. interrogans should be reported in italics. Please revise all manuscripts.
Finally English must be revised, are present some mistakes, misprints and mistakes.
Author Response
Comments and Suggestions for Authors
- Reviewer’s comment: The paper submitted by Sarma and colleagues showed the identification of protein in Leptospira interrogans by external proteome analysis. The paper is very interesting and gives important information on leptospirosis.
Author’s response: The authors thank the reviewer for their patience and time to review the manuscript, their constructive comments and appreciation.
- Reviewer’s comment: I suggest changing the title, it is too long, confused, and full of brackets.
Author’s response: The authors thank the reviewer for this important suggestion and revised the title.
- Reviewer’s comment: Need references in the sentences from line 40 to line 50. also lines 75-80. Besides in the introduction, there are too long sentences without references.
Author’s response: All necessary references are added in the revised document.
- Reviewer’s comment: Please define LC-MS/MS the first time in the text.
Author’s response: The authors added the mentioned abbreviation as well as cleared other similar issues.
- Reviewer’s comment: interrogans should be reported in italics. Please revise all manuscripts.
Author’s response: The authors gone through the entire manuscript and revised all binomials accordingly.
- Reviewer’s comment: Finally English must be revised, are present some mistakes, misprints and mistakes.
Author’s response: The authors contacted experts in English language and made a thorough revision.